# Estimates of Dietary Mineral Micronutrient Supply from Staple Cereals in Ethiopia at a District Level

**DOI:** 10.3390/nu14173469

**Published:** 2022-08-24

**Authors:** Abdu Oumer Abdu, Diriba B. Kumssa, Edward J. M. Joy, Hugo De Groote, R. Murray Lark, Martin R. Broadley, Dawd Gashu

**Affiliations:** 1Center for Food Science and Nutrition, Addis Ababa University, Addis Ababa P.O. Box 1176, Ethiopia; 2School of Biosciences, Sutton Bonington Campus, University of Nottingham, Addis Ababa LE12 5RD, UK; 3Faculty of Epidemiology and Population Health, London School of Hygiene & Tropical Medicine, Keppel Street, London WC1E 7HT, UK; 4International Maize and Wheat Improvement Center (CIMMYT), Nairobi P.O. Box 1041-00621, Kenya; 5Rothamsted Research, West Common, Harpenden AL5 2JQ, UK

**Keywords:** cereals, dietary supply, mineral micronutrients, disaggregated data, spatial data

## Abstract

Recent surveys have revealed substantial spatial variation in the micronutrient composition of cereals in Ethiopia, where a single national micronutrient concentration values for cereal grains are of limited use for estimating typical micronutrient intakes. We estimated the district-level dietary mineral supply of staple cereals, combining district-level cereal production and crop mineral composition data, assuming cereal consumption of 300 g capita^−1^ day^−1^ proportional to district-level production quantity of each cereal. We considered Barley (*Hordeum vulgare* L.), maize (*Zea mays* L.), sorghum (*Sorghum bicolor* (L.) Moench), teff (*Eragrostis tef* (Zuccagni) Trotter), and wheat (*Triticum aestivum* L.) consumption representing 93.5% of the total cereal production in the three major agrarian regions. On average, grain cereals can supply 146, 23, and 7.1 mg capita^−1^ day^−1^ of Ca, Fe, and Zn, respectively. In addition, the Se supply was 25 µg capita^−1^ day^−1^. Even at district-level, cereals differ by their mineral composition, causing a wide range of variation in their contribution to the daily dietary requirements, i.e., for an adult woman: 1–48% of Ca, 34–724% of Fe, 17–191% of Se, and 48–95% of Zn. There was considerable variability in the dietary supply of Ca, Fe, Se, and Zn from staple cereals between districts in Ethiopia.

## 1. Introduction

Mineral micronutrients are required in small amounts for normal human body growth, development, and functioning. Globally, an estimated 2 billion people have one or more micronutrient deficiencies, which is also known as “hidden hunger.” Ca, Fe, Se, and Zn deficiencies are common among populations in Sub-Saharan Africa (SSA) due to low concentrations of plant-available minerals in soils, a lack of access to or low availability of nutrient-dense foods, and poor bioavailability of minerals from cereal grains [1,2]. In Ethiopia, Zn deficiency affects 30–51% of children [3,4,5], with soil Zn deficiency being one risk factor [6]. Selenium deficiency is also widespread, with 35.5% of the population having serum Se concentrations below the threshold of adequacy (70 µg L^−1^), based on nationally representative surveys of serum micronutrient concentrations [7]. Within the Amhara region (in northern Ethiopia), 49.1% of young children had Se deficiency, with a higher burden in western Amhara (91%), while there was little or no deficiency in children from the eastern part of the region [8]. In addition, Fe deficiency has been reported among 17.8% of preschool children, 9.1% of school-aged children, and 10% of women of reproductive age (WRA) [3].

Studies based on dietary intake estimates also show the public health significance of micronutrient deficiencies. The 2011 Ethiopian food consumption survey indicated that, nationally, about 13% and 50% of WRA had inadequate Fe and Zn intake, respectively [9]. For Ca, 89%, 91%, and 96% of WRA, pregnant women, and adolescent girls in Ethiopia, respectively, had dietary intakes below the Estimated Average Requirements (EARs) [10]. However, the intake of these micronutrients varied significantly among regions. For example, the prevalence of inadequate Fe intake among women ranged from 6% in the Amhara region to 84% in the Somali region. In addition, intakes are often higher among urban populations than in rural residents [9]. Emerging evidence of the importance of spatial variation in the mineral concentration of staple cereals and human blood indicates that the national mean, as often reported in the food composition data or food consumption survey, is not a good representation of dietary mineral supply. This warrants studies based on spatially disaggregated estimates of dietary mineral intakes and estimates of inadequacy. This approach can be used as a basis to prioritize and design localized interventions for greater efficiency.

Cereals contribute significantly to daily nutrient intake, particularly in developing countries where they typically account for >50% of the daily calorie intake [11,12]. Similarly, in Ethiopia, on average, cereals account for >65% of daily energy consumption, with maize, sorghum, teff, and wheat being the major contributors [13]. The national food consumption survey also reported that cereals contribute to more than 70%, 60%, and 59% of the total dietary consumption among adults in the Amhara, Oromia, and Tigray regions, respectively [9]. In Ethiopia, diets are typically plant-based with very low consumption of animal-source foods [14]. Cereal crops are typically produced, processed, and consumed locally, suggesting that the quantity of local total crop production could be a useful proxy indicator for consumption.

A recent survey was conducted as part of the GeoNutrition project, in which cereal grains were sampled across the regions known for cereal production (91% of total national cereal production) in Ethiopia from Oromia (51%), Amhara (32%), and Tigray (8%) regions [15]. There was substantial variation in Ca, Fe, Se, and Zn concentrations between crop types (including barley, maize, sorghum, teff, and wheat). In addition, the geographical location associated with soil and geological factors was a main determining factor for the variability of mineral concentrations in cereal grains, at distances of over 250 km [16]. The observed variation in grain Se concentrations was consistent with human blood data, including in the Amhara region, which displayed a strong east-west gradient in Se concentrations in grains and blood plasma [17]. Previous studies have estimated dietary mineral supplies in Ethiopia using single national values of grain composition [9,18], often in combination with national food supply data [18,19,20,21]. A major limitation of these studies is the failure to capture sub-national variability in cereal consumption and/or lack of consideration of the presence of between or within species variability in the mineral composition of cereal grains. This limits the ability to design spatially targeted interventions, including agronomic biofortification of staple crops. The present study aims to estimate the dietary supply of Ca, Fe, Se, and Zn from major cereals at the district level in Tigray, Amhara, and Oromia regions, which represent the major agrarian regions in Ethiopia.

## 2. Materials and Methods

### 2.1. Study Settings

Over 114 million people live in Ethiopia [22], with the majority (80%) of the population relying on agriculture for employment and income. The agricultural system is characterized by diversified crop production. Cereals account for 80.7% (10.2 of 12.6 million hectares of cultivated land) of the total cropland and contribute up to 87.5% share of the total agricultural production. In the year 2020 and 2021, teff, maize, wheat, sorghum, and barley covered 28%, 24%, 18%, 16.5%, and 9% of the total cropping area, respectively, and contribute >90% of the overall crop production [23]. In addition, teff (15.1%), wheat (12.5%), maize (9.3%), sorghum (42.4%), and barley (10%) contribute 89.3% of the total cereal crop production tonnage. Oromia (45.3%), Amhara (35.3%), and Tigray (7.4%) are the major cereal-producing regions, accounting for >88% of the total cropping area and total production, respectively [24]. Barley, maize, sorghum, teff, and wheat consumption were considered in the present analysis.

### 2.2. Crop Production and Cereal-grain Mineral Composition Data

District-level cereal production data were obtained from the agricultural sample survey conducted by the Central Statistical Agency of Ethiopia (CSA) [25]. The production quantity of each cereal, i.e., barley (*Hordeum vulgare* L.), maize (*Zea mays* L.), sorghum (*Sorghum bicolor* (L.) Moench), teff (*Eragrostis tef* (Zuccagni) Trotter), and wheat (*Triticum aestivum* L.) was extracted for 326 districts for the year 2017 and 2018, and the proportion of each cereal for total cereal production was calculated for each district. The proportions were multiplied by 300 g (0.3 kg) [17] to provide a standardized per capita estimate of cereal consumption for adult women.

Cereal grain Ca, Fe, Se, and Zn composition data (n = 1386) were obtained from the GeoNutrition survey as reported previously [16]. The GeoNutrition grain samples were collected from the Tigray, Oromia, and Amhara regions between 2018 and 2019. The sampling frame consisted of agricultural lands being cultivated for crop production based on high-resolution satellite images. Sample locations were distributed across the sampling frame for spatial balance. Samples of cereal grains were taken from mature standing crops, crop stacks, or household stores. The samples were milled, acid digested, and the mineral concertation was analyzed using Inductively Coupled Mass Spectroscopy (ICP-MS; iCAP-Q: Thermo Fisher Scientific, Bremen, Germany), which allows the detection of small concentrations of minerals in a sample. The detailed procedure for sample collection, preparation, and analysis is reported elsewhere [16].

Grain mineral concentration was estimated at district-level by crop type using block kriging (for Ca and Zn) and conditional simulation after Gaussian transformation (for Fe and Se). The maize Zn and teff Ca concentration data had outliers, and the block mean was estimated with a 95% confidence interval, which is suitable to handle predictions from skewed data. For barley and sorghum, ordinary kriging with and without log transformation was applied to derive the district block mean mineral concentrations.

### 2.3. Estimation of Dietary Mineral Supplies and Contribution to Requirements

The district-level cereal consumption data was merged with district-level crop mineral composition data to calculate cereal mineral supply (Ca, Fe, Se, and Zn). Estimates of dietary mineral supplies were compared against a threshold dietary recommendation value for WRA (there was no particular interest in using WRA dietary recommendation values), to illustrate the potential nutritional implications of district-level variation in dietary mineral supplies from cereals. Cereals are known to contain antinutritional factors such as phytate that reduce mineral bioavailability. Thus, an Estimated Average Requirement (EAR) value for WRA (aged 18–24 years) considering low mineral bioavailability (phytate to Zn molar ratio > 15) was applied to estimate the dietary contribution of staple cereals to daily Ca, Fe, Se, and Zn requirements [26,27]. The EAR is the usual daily intake of a nutrient required to meet the needs of half of healthy individuals in an age-and sex-specific population group. The EAR values for WRA are: 860 mg for Ca, 22.4 mg for Fe, 45 μg for Se, and 10.2 mg for Zn [27].

Geospatial analysis and thematic mapping of Ca, Fe, Se, and Zn supply from staple cereals and their contribution to dietary requirements were conducted using ArcGIS 10.8 (Version 10.8, ESRI, Redlands, CA, USA). Descriptive statistical data analysis was performed in Microsoft Excel 2016 and Stata SE, version 14 (StataCorp, 4905 Lakeway Drive, College Station, TX 77845, USA).

## 3. Results

### 3.1. District-Level Cereal Production

Staple cereal production showed wide district-level variation in Ethiopia, with a total of 7.57 million tons of staple cereal produced in the Amhara, Oromia, and Tigray regions. Among these, maize had the largest share of total cereal production (32.1%), followed by teff (20.3%), sorghum (18.8%), wheat (17.5%), and barley (11.3%). Maize was the dominant crop in approximately 31% of the districts, compared to 24%, 19%, 16%, and 10% of sorghum, teff, wheat, and barley, respectively.

### 3.2. Grain Mineral Composition

The number of cereal samples from the GeoNutrition survey considered in the present study were teff (n = 371), wheat (n = 326), maize (n = 301), sorghum (n = 138), and barley (n = 181). The grain mineral composition showed variation by crop type and location. Overall, teff had the greatest mineral concentrations, while maize had the lowest. District-level estimates of maize grain Ca concentration ranged from 19 to 21 mg kg^−1^. In addition, district-level estimates of wheat grain Zn concentration ranged from 26 to 30 mg kg^−1^ (Table 1). There was substantial variation in cereal grain composition between districts as generated by the block kriging and conditional simulation methods (Appendix A).

### 3.3. Dietary Mineral Supply from Cereals

Dietary mineral supplies from staple cereal grains were estimated at the district level, based on consumption of 300 g capita^−1^ day^−1^ on a dry basis for an adult woman. Across districts, the median supply of cereal grains was 146 mg of Ca, 23 (16.8–34.1) mg of Fe, 25.1 (18.2–36.4) µg of Se, and 7.1 (6.4–8.0) mg capita^−1^ day^−1^ of Zn. However, there was substantial variation between districts, as clearly indicated in Table 2, Figure 1, and Table 1. For example, district-level estimates of dietary Ca supply from cereals ranged from 83 to 201 mg capita ^−1^ day^−1^ while district-level estimates of dietary Zn supply from cereals ranged from 6.4 to 8.0 mg capita^−1^ day^−1^ (Table 2).

At district level, cereals can provide a median of 17% Ca, 103% Fe, 56% Se, and 69% Zn to the daily EAR. However, the contribution of cereals to the EAR varied considerably between districts (Table 3; Figure 1).

For the majority of districts, cereals supplied less than one-fifth of the requirements for Ca, the lowest of the minerals studied. A low dietary supply of Ca from cereals was estimated to occur across the country. However, a greater supply of Ca was observed in teff-consuming districts. Similarly, a relatively greater dietary Ca consumption was predicted in wheat-producing areas (Figure 1). Estimated supplies of Fe from cereals showed a strong regional pattern, with higher supplies in the Tigray region and parts of southern Oromia (Figure 1). Again, the higher Fe supply was driven by teff consumption. Some districts in central Amhara had estimated Fe supplies from cereals above the tolerable upper-intake level (>45 mg per day) (Appendix A), although it is likely that a low proportion of this Fe is bioavailable.

Districts with low dietary Se supplies from cereals (<50% of the daily EAR) were concentrated in the northwestern parts of Tigray region, the western parts of the Amhara region, and the western Oromia (Figure 1). Moderate Se supply was observed in the central and southeastern parts of the Oromia region. However, there were some areas along the Rift Valley where estimates of dietary Se supply reached up to 190 μg capita^−1^ day^−1^, substantially above the EAR, yet below the tolerable upper-intake level (400 μg capita^−1^ day^−1^). District variation in dietary Se supply estimates was driven mainly by variation in crop composition driven by soil grain concentration and availability of grains.

In the present study, dietary Zn supply from cereals was lowest in the central Amhara, western Tigray, and eastern and western Oromia, but greater supply was observed in parts of the southern and central Oromia and Tigray regions (Figure 1). The districts with the greatest estimated dietary Zn supplies were found where predicted district-level crop Zn concentrations were greatest. Cereals supplied more than half of the requirements in the majority of districts, although it is likely that a low proportion of this Zn from cereals is bioavailable.

## 4. Discussion

The contribution of staple cereals to daily mineral micronutrient supplies was estimated at district-level in Ethiopia. Across districts, daily consumption of 300 g day^−1^ of staple cereals contributed about 17% of Ca, 103% of Fe, 56% of Se, and 70% of Zn of the EAR for an adult woman. However, there were significant sub-national and between-district variations, mainly driven by differences in the types of cereal available for consumption and mineral grain concentrations. These disaggregated estimates reveal areas of the country with potentially suboptimal mineral micronutrient intake and higher risks of mineral deficiencies. These include districts in western Tigray and western and eastern Oromia for Ca; districts in western, central, and eastern Oromia for Fe; districts in western Amhara and Oromia; districts in north-eastern Amhara; and western and eastern Oromia for Zn.

Dietary mineral deficiencies are widespread in the SSA [21]. This is largely due to the consumption of plant-based diets, but with low consumption of animal-source foods [2]. Plant-based diets typically contain a low concentration of mineral micronutrients. On the contrary, plant foods contain antinutritional factors like phytate, polyphenols, and dietary fibers that negatively affect mineral bioavailability [28]. In Ethiopia, nationally, cereals provide approximately 64% of the daily energy intake, with even greater contributions among the rural population. Barley, maize, sorghum, teff, and wheat are the major cereals contributing to the daily nutrient intake [13].

In the present study, staple cereals from major parts of the country are limited in their ability to supply adequate mineral micronutrients to satisfy the daily average requirements. Previous studies also reported similar findings, drawing on national food consumption survey data (NFCS) [9] and the national-level Food Balance Sheet, combined with food composition tables [18].

For an adult, a daily intake of 800 mg day^−1^ of Ca allows maintenance of optimal bone Ca balance and reduces the risk of Ca deficiency [29]. However, in the present study, the estimated average dietary Ca supply from staple cereals was only 146 mg day^−1^, less than one-fifth of the daily requirement. Studies reporting higher Ca from non-fortified cereal grains are available. For example, a study reported that 300 mg of the daily Ca supply comes from cereal grains, accounting for 60% of the total dietary Ca supply [10]. The present study considered only five staple cereals. However, a higher daily dietary intake of Ca could be obtained from common food items such as milk, finger millets, and water consumption depending on the hardness of the water [16,30,31], which were not considered in the present study. The daily consumption of additional foods rich in Ca could contribute to the daily Ca intake and reduce the risks of inadequate dietary Ca intake compared to the one estimated in this study.

In the present study, the median Fe supply from staple cereals (23 mg day ^−1^) was above the EAR for adult women (i.e., 22.4 mg day ^−1^) [27]. A dietary estimate using the Ethiopia NFCS data also indicated that existing dietary consumption provides about 27–37 mg of Fe capita^−1^ day^−1^ [9], which is close to our estimate. Compared to other resource-poor countries [32,33,34], iron deficiency anemia is less of a severe public health problem in Ethiopia, where 17.8% of preschool children, 9.1% of school-aged children, and 10% of WRA had low storage iron as measured by ferritin [3].

This may be due to exogenous Fe contamination of cereal grains during the harvesting and threshing of cereals [35], combined with the tendency to consume fermented injera flatbread, which increases bioavailability [36]. Analysis of grain samples from stores shows that extrinsic iron could reach up to 74% of the total iron in teff, 66% in sorghum, 65% in barley, and 45% in maize and wheat. However, Fe from extrinsic sources has limited bioavailability [37,38].

In the present study, cereals could contribute a median of 25.1 μg day^−1^ (Q_1_, Q_3_: 18.2, 36.4 μg day^−1^) of Se to the daily requirements. Among the cereals, teff and wheat could supply a larger proportion of the daily Se supply compared to other cereals. This is the first estimate of Se supply from cereals where the Se intake was not included in the recent NFCS [9]. A study from Northwestern Ethiopia indicated that 49.1% [8] of young children were Se deficient (serum Se below 70 µg L^−1^). In addition, a national survey based on Se biomarkers showed that 35.5% of children were Se deficient, with a marked spatial variation over the country, where Se status was better in the northeast and eastern parts of Ethiopia. Western parts of the country with better crop production [25] had inadequate Se status [7], and the grain Se concentration was poor in these areas [17]. The Se concentration in soil and crops varies significantly across Ethiopia [17,39,40]. This could influence the contribution of cereals to the daily Se supply and, ultimately, the risk of Se deficiency at a population level. Consistent with human blood data, in the present study, low Se supply from cereals was observed in the northwestern and western parts of the country. Moreover, mineral deficiency risk assessment based on food supply data estimates that dietary consumption supplies 36 to 45 µg day^−1^ of Se and the risk of Se deficiency reaches in the range of 26–50% [21].

A report of the NFCS shows the daily average Zn intake (1.7 mg for children and 7.2 mg for adult women) [9,41] is far below the daily requirement (EAR of 10.2 mg day ^−1^ for adults and 8.3 mg day^−1^) [27,42]. Another study based on national-level food balance sheet reported a dietary supply of potentially absorbable Zn of only 2.7 mg person^−1^ day^−1^ [18] compared to the daily requirement of 10.2 mg day^−1^ [27]. Low Zn supplies from cereals could be due to low soil Zn content or low phytoavailability of Zn, resulting in a low Zn concentration in grains [43,44]. However, the high phytate content (phytate to Zn ratio > 15) further decreases Zn bioavailability and increases the risk of Zn deficiency in humans [6,45]. Traditional cereal-processing methods such as fermentation, germination, soaking, and roasting reduce phytate content and increase mineral bioavailability [46]. According to a study based on nationally representative biomarker data, Zn deficiency affects approximately 72% of the Ethiopian population across all demographic groups. WRA in most parts of the country (except central parts of Ethiopia and other small parts of northern and eastern Ethiopia) is virtually certain to be Zn deficient [47]. Zn deficiency indicated by serum Zn concentrations is spatially dependent over short distances but without a clear geographical pattern. The Ethiopian NFCS indicated that the daily Zn intake is about 7.2 mg day^−1^, which is very close to our estimate [9]. This widespread, low dietary contribution of cereals to the daily Se and Zn supply warrants interventions that support nutrient-dense cereal production [48,49].

Subnational disaggregation of dietary mineral supply estimates is important in nations where food systems are localized and there is spatial variability in food consumption patterns and/or crop composition. Subnational data may provide more accurate estimates of dietary mineral intakes and risks of deficiency. This can be used to inform location-oriented interventions, including agronomic biofortification through the application of micronutrient fertilizers. Several countries are developing national, sub-national, and district-level soil database dissemination platforms that contain information from the field and laboratories to help make informed use of soil resources and inform fertilizer application. The Ethiopian Soil Information Service (EthioSIS) project under the Agriculture Transformation Agency in the Ministry of Agriculture and Natural Resources is developing a soil database [50]. Among other benefits, soil mapping is helpful to inform local level recommendations for the application of mineral blended fertilizer applications for improved crop yield and the production of nutrient-dense staple crops, which in turn may contribute to alleviating micronutrient deficiencies [50,51].

The nutritional quality of cereals exhibits spatial variation at scales relevant for crop production and consumption patterns. This spatial variation can be attributed to several factors, including soil physicochemical characteristics, climatic factors, and differences in the physiology of the crops [52]. For example, the mineral micronutrient content of staple cereal grains in Ethiopia was associated with soil and environmental covariates including soil pH, soil organic matter, temperature, rainfall, and topography. Such is the scale of spatial variation in crop quality that the location of residence is likely to be a strong influencing factor in determining the dietary intake of micronutrients for populations [16,17]. The influence of soil and environmental factors may be different for the same mineral micronutrient in a specific cereal. For example, soil pH positively influenced grain Se concentration for maize, sorghum, teff, and wheat, but was negatively correlated with teff Zn concentration. In addition, soil organic carbon was negatively correlated with wheat grain Se concentration but positively correlated with the Zn concentration of wheat grain [16].

The findings of the present study should be considered in light of some limitations. Only major cereals such as maize, wheat, teff, sorghum, and barley were considered in the mineral supply estimation, neglecting minor cereals and other food crops. The decision to limit the analysis to mineral supplies from major cereal crops was made due to the availability of crop composition data. The analysis covered the major cereal-producing regions of Ethiopia and would have limited value in predominantly pastoralist areas. In addition, the use of crop-production data as a proxy for consumption neglects the significance of the influx of cereals from other areas into the local market. The approach assumes that consumption patterns mirror production patterns, and the available data do not allow further disaggregation by factors that potentially influence consumption, such as socioeconomic status. It is also important to note that the consumption and utilization of minerals could be different from the present supply estimates because of factors affecting bioavailability, food consumption patterns, food processing, and food preparation techniques.

## 5. Conclusions

The present study reveals the presence of sub-national and district-level spatial variation of mineral supplies from the major cereals in Ethiopia. Between-district variations in mineral micronutrient supply occur mainly due to geographical influence, the type of cereals available for consumption, and the crop mineral composition of grains. This demonstrates the value of subnational disaggregated data on food systems and population micronutrient status to inform the design and evaluation of interventions to alleviate mineral micronutrient deficiencies. Further studies are warranted to estimate dietary mineral supplies of other food items as well.

## Figures and Tables

**Figure 1 nutrients-14-03469-f001:**
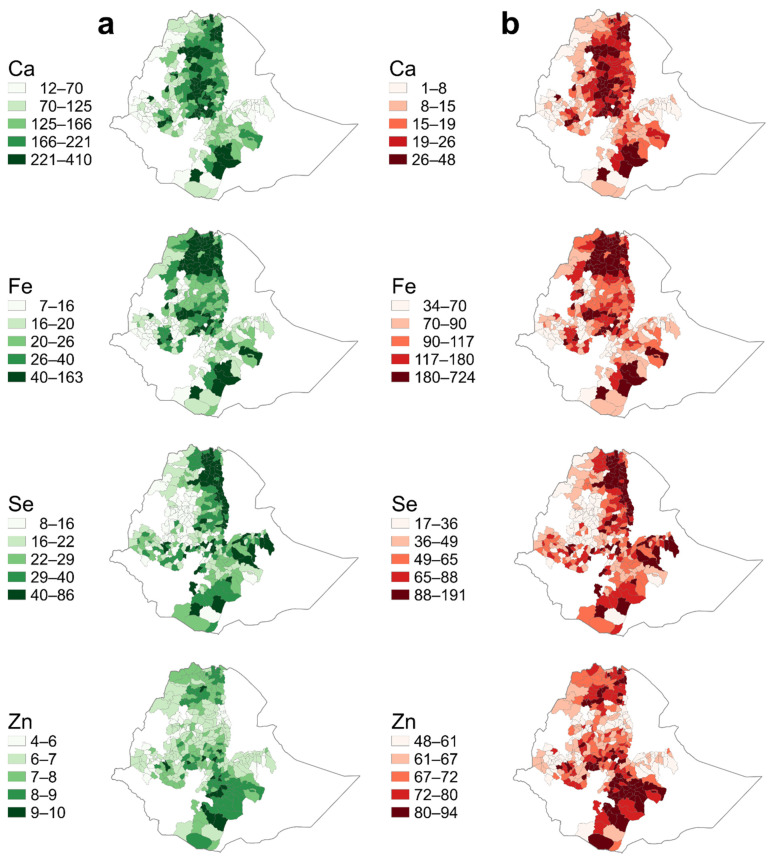
District-level estimates of dietary calcium (Ca), iron (Fe), selenium (Se), and zinc (Zn) supplies from staple cereals (barley, maize, sorghum, teff, and wheat), and contributions to the Estimated Average Requirement of Women of Reproductive Age in Ethiopia. (**a**) District level Ca, Fe, Se, and Zn supply from cereals assuming 300 g capita^−1^ day^−1^ total cereal consumption, with mixture of cereal crops in proportion to crop production quantity in a district, (**b**) District-level percentage contribution of cereals to Ca, Fe, Se, and Zn EAR for WRA. The daily Ca, Fe, and Zn supplies are expressed in mg capita^−1^ day^−1^ and Se is in µg capita^−1^ day^−1^.

**Table 1 nutrients-14-03469-t001:** District-level mineral micronutrient concentrations (mg kg^−1^) in cereal grains in Ethiopia.

Grain Type	Ca	Fe	Se	Zn
Barley	403.0 (68–437)	60.0 (51–76)	25.0 (20–29)	30.0 (27–34)
Maize	77.0 (62–85)	24.0 (23–25)	15.0 (8–22)	20.0 (19–21)
Sorghum	191.0 (165–243)	59.0 (53–72)	12.0 (8–18)	20.0 (19–21)
Teff	1575.0 (1485–1611)	270.0 (199–340)	61.0 (42–93)	29.0 (28–31)
Wheat	427.0 (381–467)	45.0 (43–48)	8.0 (6–10)	28.0 (26–30)

Note: the concentrations are expressed as median (first and third interquartile). Units are mg kg^−1^ for Ca, Fe, and Zn; μg kg^−1^ for Se.

**Table 2 nutrients-14-03469-t002:** District-level dietary mineral micronutrient supplies (mg capita^−1^ day^−1^) from staple cereals for adult women in Ethiopia.

Grains	Ca	Fe	Se	Zn
Teff	83.0 (29–144)	10.9 (4.8–24)	5.6 (2.0–13.1)	1.50 (0.54–2.60)
Wheat	14.0 (3.0–30.0)	1.4 (0.3–3.3)	2.0 (0.5–5.5)	0.88 (0.19–1.89)
Maize	4.5 (1.5–9.8)	1.4 (0.5–3.4)	3.4 (1.3–7.4)	1.20 (0.39–2.66)
Sorghum	8.3 (2.1–22.0)	3.0 (0.7–7.0)	4.6 (1.2–11.2)	0.82 (0.23–2.10)
Barley	5.0 (1.8–20.7)	0.8 (0.3–3.7)	0.3 (0.1–1.3)	0.36 (0.12–1.51)
Cereals ^‡^	146.0 (83–201)	23.0 (16.8–34.1)	25.1 (18.2–36.4)	7.1 (6.4–8.0)

Note: The mineral supplies are expressed in median (first and third quartiles); units are mg kg^−1^ for Ca, Fe, and Zn; μg kg^−1^ for Se; ^‡^ indicates the district-level median mineral micronutrient supply from staple cereals (barley, maize, sorghum, teff, and wheat). Since we used the median as a measure of central tendency indicating positional average rather than numerical average, the sum of median mineral supply from each cereal is not equivalent to the aggregated median supply from staple cereals.

**Table 3 nutrients-14-03469-t003:** Median district-level dietary contribution (%) of staple cereals to mineral micronutrient estimated average requirement for adult women in Ethiopia.

Cereals	Ca	Fe	Se	Zn
Maize	0.5 (0.2–1.1)	6.8 (2.1–15.3)	7.5 (2.8–16.4)	12.0 (3.8–26.1)
Wheat	1.6 (0.3–3.4)	6.3 (1.5–14.8)	44.4 (12–123)	8.6 (1.9–18.5)
Teff	9.7 (3.3–16.7)	48.5 (21.6–107)	12.3 (4.5–29.1)	14.7 (5.3–25.5)
Sorghum	1.0 (0.2–2.6)	13.2 (3.2–31.2)	10.0 (2.7–25.0)	8.0 (2.2–20.3)
Barley	0.6 (0.2–2.4)	3.6 (1.3–16.6)	0.7 (0.2–2.9)	3.6 (1.2–14.8)
Total ^†^	17.0 (0.0–23.0)	103.0 (75.0–152.0)	56.0 (40.0–81.0)	69.0 (62.0–78.0)

Note: The district-level dietary contribution (%) of staple cereals is expressed in median (first and third interquartile). Units are mg kg^−1^ for Ca, Fe, and Zn; μg kg^−1^ for Se. ^†^ Refers to the median dietary contribution of the five staple cereals to the daily EAR of adult women, not the numerical sum of the median contribution from each cereal.

## Data Availability

The data presented in this study are available within the Appendix A.

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
