# Peer review of "Estimates of Dietary Mineral Micronutrient Supply from Staple Cereals in Ethiopia at a District Level"

_nutrients, 2022, doi:10.3390/nu14173469_

Round 1
Reviewer 1 Report
Comments and queries are presented in the article.

Author Response
Hello dear reviewer,
Thanks for having your valuable imputs and suggestions for this study. We replied to the comments within the PDF attachd and we also made sbstantial editions and revisions in the revised manuscript.
If there are queries, we would be happy to respond.
Kind regards
Abdu

Reviewer 2 Report
Major concerns
1) It is assumed that the Ethiopian population feeds exclusively on cereals produced in Ethiopia. Is there no import of cereals (e.g. wheat) from other countries?
2) The estimate of 300g / day per capita of cereals consumed seems very imprecise. Moreover, the micronutrient content in crops change according to the amount of rainfall, the type and quantity of fertilizers applied and other environmental conditions.
3) The estimated reduction of mineral absorption by phytate seems to me rather arbitrary.
4) A considerable amount of Calcium can result from the consumption of water rich in calcium. This variable was not considered by the authors.
In conclusion, this study focuses on two aspects. The mineral content in the various cereal crops, geographically scattered throughout the country and the content of the same in the diet of the population. Both of these aspects, however, have been treated in a superficial way that does not allow to draw the conclusions described, nor can Authors explain with their results the micronational deficiencies highlighted by epidemiology.
Author Response
Hello dear reviewer,
Thanks for having your concerns. We have addressed the suggestion and explained concerns that need further explanation in detail. We made the relevant thorough editions to the English language by profession as indicated in the revised version. The detail point-point responses are detailed under.
With kind regards
Abdu
